# Long-Term Changes in Vertical Jump, H:Q Ratio and Interlimb Asymmetries in Young Female Volleyball Athletes

**DOI:** 10.3390/ijerph192416420

**Published:** 2022-12-07

**Authors:** Cesar Cavinato Cal Abad, Marcos Winicius Rodrigues Lopes, Jerusa Petróvna Resende Lara, Anderson Jose Santana Oliveira, Raphael Planas Correa da Silva, Elder Aparecido Facin, Antonio Jose Izar, Fabiano Gomes Teixeira

**Affiliations:** Reference Centre of Sport Science of Social Service of Industry, CRCE-SESI, São Paulo 05574-001, Brazil

**Keywords:** injury, isokinetic dynamometer, performance, physical fitness, sport, training, vertical jump

## Abstract

The present study aimed to examine the changes that occurred in vertical jump and isokinetic dynamometer (ISK) performances at the beginning of a preparatory period (PRE) and at the start of the competitive period (POST). Sixteen U-17 elite female volleyball players, from a national level (15.34 ± 1.19 years; 66.35 ± 7.95 kg; 169.22 ± 24.79 cm), performed bilateral squat jump (SJ), bilateral and unilateral countermovement jump (CMJ) and unilateral ISK tests for knee flexors (Fl) and extensors (Ex) both at 60°/s and at 300°/s. Peak torque (PT) and the hamstring-to-quadriceps (H:Q) ratio were assessed by concentric actions. Asymmetries were calculated by the percentage differences between dominant (DOM) and nondominant legs (NDOM). The paired Student’s *t*-test was used for comparisons at a level of significance of 5%. The effect size was also calculated. A significant increase was found for both SJ (15%; *p* = 0.004; ES = 0.82) and CMJ (12%; *p* = 0.017; ES = 0.62). The PT of NDOM flexors at 60°/s was significantly lower than DOM both at PRE (4.6%; *p* = 0.048; ES = −0.22) and POST (6.3%; *p* = 0.037; ES = −0.33). The NDOM extensors at 60°/s had a significantly lower PT than DOM at POST (7.0%; *p* = 0.048; ES = −0.23). Both DOM and NDOM flexors at 60°/s had a PT enhancement at POST related to PRE (6.7%; *p* = 0.031; ES = 0.51 and 5.6%; *p* = 0.037; ES = 0.48, respectively). The PT of NDOM extensors at 300°/s increased at POST in comparison to PRE (7.9%; *p* = 0.038; ES = 0.27). The NDOM at 300°/s had a H:Q ratio higher than DOM both in PRE and POST (8.6%; *p* = 0.041; ES = 0.37 and 11.6%; *p* = 0.013; ES = 0.71, respectively), and the highest H:Q ratios were lower than the reference values (<80%). The asymmetry of the unilateral CMJ was higher at POST than at PRE (102%; *p* = 0.03; ES = 0.81). The PT for the flexors at 300°/s and the H:Q ratio at POST exceeded 10%. In conclusion, a training program of 15 weeks increased the neuromuscular performance of young volleyball athletes, but many H:Q ratios and asymmetries remained out of the normal recommendation. Volleyball professionals should carefully apply an adequate training program to enhance physical fitness performance without increasing the risk of lower limb injuries concurrently.

## 1. Introduction

Volleyball is defined as an intermittent court sport that requires players to compete in frequent short bouts of high-intensity exercise followed by periods of low-intensity activity [1]. It requires interchanged energy from both aerobic and anaerobic metabolism [2], and many technical skills (serves, reception, digs, spikes and blocks) and physical fitness (sprinting, agility, power and strength) are related to player success [3,4].

In addition to technical and tactical skills, the ability to jump high has also been considered an essential skill related to volleyball performance [5]. Per game, women’s volleyball athletes may execute close to 36 jumps, with more than 50% of defensive landings occurring only on one foot [6]. These relatively high numbers of unilateral actions executed during trainings and matches could lead to a loss of muscle balance, an increase in muscle asymmetries and, subsequently, in some injury risk factors.

Strength training focused on hip flexion and abduction, hamstring, core, and abdominal musculatures to aid in proper lower extremity alignment and muscle recruitment patterns has been considered a good tool to prevent injuries, develop muscle strength and improve physical fitness related to volleyball performance [7,8]. It is particularly important for young players because injury may be detrimental to the training process, volleyball performance and the athlete’s career. [9].

Muscular imbalances and bilateral strength asymmetry have been suggested as important risk factors for lower limb injuries [10,11]. Although it is difficult to generalize, the normal muscular balance for the hamstring-to-quadriceps (H:Q) ratio has been considered to be 50% to 80% [12,13]. For asymmetry, values greater than 10–15% have been related to a higher risk of injury in soft tissues, joints and muscles [12,13,14,15,16].

Asymmetries have been commonly assessed by vertical jump (VJ) and isokinetic peak torque for flexion and extension in the knee joint for both dominant (DOM) and nondominant (NDOM) limbs in many team sports athletes. However, the absolute cut-off values for both the H-Q ratio and interlimb asymmetries remain relatively contradictory. While many studies report poor or no relationships among ISK performance, VJ height, asymmetries and injuries [17,18,19], many others suggest important and significant relationships between asymmetry, poor performance and injury risk in athletes [20,21,22].

In addition to the contradictory findings of the aforementioned studies, most of them investigated adult players and had short-term or transversal designs. Thus, the long-term training effect on VJ performance, H:Q ratio, and interlimb asymmetries during a season remains under described in young female volleyball athletes. This information would be important for sport staff (i.e., sports scientists, managers, physiotherapists, coaches, and physical conditioners) because they could more adequately manage the facilities and the training contents to prevent injuries and to optimize the overall sports performance of young volleyball players.

With this in mind, the aim of the present study was to examine the long-term changes in muscle imbalances, asymmetries and VJ performances of young female volleyball athletes at the start of a preparatory period (PRE) and at the start of a competitive period (POST) through a season. We hypothesized that a long-term training program comprised of general and specific volleyball training contents could improve VJ performance and decrease some muscle imbalances and interlimb asymmetries. We also expected that asymmetries would vary according to each physical test (VJ and ISK), muscle type (knee flexors or extensors) and angular velocities (60°/s and 300°/s).

## 2. Materials and Methods

### 2.1. Study Design and Overall Procedures

The present study has descriptive characteristics with quasi-experimental and longitudinal designs. Prior to the study, the volunteers and their parents or guardians were informed of the procedures to be followed and the measurements to be taken, as well as the possible risks and benefits. The athletes and their guardians read and signed the informed consent form agreeing to participate in the study. All of the volunteers could withdraw from the experiment at any time without any prejudice. The study was submitted and approved by the local ethics committee. All procedures were performed in accordance with the Helsinki Declaration.

### 2.2. Participants

Only the players who completed at least 90% of all the planned training sessions composed our sample. Overall, 16 national-level elite U-17 female volleyball players (15.34 ± 1.19 years; 66.35 ± 7.95 kg; 169.22 ± 24.79 cm) volunteered to participate in the study. All the players had their onset of menarche at least two years before the study, and at this age, their muscle mass and maximum strength performance remained in development and sensitive to training. The players were relatively experienced athletes with a time of volleyball practice of 4.8 ± 1.7 years. They were champions of the most important Brazilian tournament at the state level and placed in the top 3 squads at the national level. All of them were experienced with strength training for at least three years. The players were also familiarized with the protocol of the study, showed a good health condition and had no previous serious lower limb injuries at least six months before the study. Then, they were heated, and the evaluator explained the dynamics of the evaluations and gave the guidelines for each evaluation.

### 2.3. Experimental Procedures and Measures

Initially, body mass and body height were measured. The sequence of evaluations was SJ, CMJ, and ISK. The evaluations occurred at the beginning of the preparatory period (PRE) and at the start of a competitive season (POST). The PRE and POST periods were interspersed by 15 weeks. During this period, the athletes had a full competitive schedule comprising their daily routines, which included training and matches.

### 2.4. Training Schedule

The training schedule was from February (PRE) to May (POST) with a preparatory and a competitive period totalling 15 weeks (Table 1).

Overall, there were 67 training sessions at the gym and 72 training sessions at the volleyball court. The training sessions were six days per week (from Monday to Saturday), occurring always in the afternoon (from 1:30 pm to 6:30 pm) and ranging from 90 to 180 min. On-court volleyball training focused on volleyball skill development, as well as learning and improving tactical elements of the game. The training program is described in Table 2.

On week 13, the athletes started the competitive phase with an official match, and on week 14, they had two more official matches. All games were at home, and before each one, the athletes retained their routines related to nutrition, hydration, sleep and rest.

### 2.5. Anthropometric Measures

All anthropometric measurements were assessed using the International Society for the Advancement of Kinanthropometry (ISAK) standardized protocol [23]. For body mass, an electronic scale (Glass 200 G-Tech, Zhongshan, China) was used, and the measurements were recorded with an accuracy of 0.1 kg. The players were barefoot and dressed in shorts and a T-shirt. Body height was measured by a stadiometer close to 0.01 m (Sanny ES2020, São Bernardo do Campo, Brazil). The players assumed the orthostatic position of the body, feet together and eyes on the horizon [23].

### 2.6. Vertical Jumping Ability

The bilateral SJ and CMJ and the single leg (unilateral) CMJ were the VJ jump tests assessed. Prior to the VJ tests, the participants executed a warm-up comprising 5–10 min of moderate cycling (5–6 of RPE 0–10 scale) followed by general mobility exercises and ballistic stretching. Before the VJ, the athletes performed three SJ and three CMJ at submaximal intensities interspersed by 30 s. After 5 min of recovery, the athletes performed the attempts for the VJ tests. The vertical jump heights of the squat jump (SJ) and countermovement jumps (CMJ) were investigated using an infrared device (OptoJump, Microgate, Italy). In SJ, players were instructed to stand, bend their knees to approximately 90°, and jump. Participants had to avoid any countermovement as much as possible and were instructed to stop for 2 s in each phase. In the CMJ, participants were instructed to stand, lower themselves into a self-selected knee flexion, and immediately jump. The arms were placed on the hips in both SJ and CMJ tests. Participants were instructed to avoid any knee flexion prior to landing on the SJ and CMJ, and the evaluator visually verified the results. Three attempts were made for each jump, and the best result was entered in the data analysis. To ensure the correct form of performing unilateral CMJ for both DOM and NDOM legs, the participants started from a static position standing on one foot and were instructed to perform a countermovement (descent phase), followed by a rapid and vigorous extension of the lower limb joints. The hands remained on the hips, and the athletes were instructed to jump as high as possible [18,20]. These tests have good reliability and reproducibility [24,25].

### 2.7. Concentric Peak Torque

Prior to the ISK test, the participants repeated the same general standardized warm-up as the VJ tests followed by a 5 submaximal repetition of concentric-concentric flexion-extension movements and a 5 min of a complete passive recovery. The players were seated in an upright position on the bench with the trunk at 90°. The trunk and shoulders were secured by two belts, and the tested leg was attached to the thigh support strap. The distal part of the leg was secured to the adjustable lever arm by three-way padded cuff fingers above the lateral malleolus. The axis of rotation of the dynamometer was aligned with the lateral femoral condyle. The untested leg was left to move freely. Passive assessment of gravitational torque was determined for each subject. The lever arm was placed closer to the horizontal position without hamstring muscle tension. The peak torque (PT) expressed in Newton meters (N·m^−1^) was assessed concentrically for both knee extensor and flexor muscles in both the DOM and NDOM legs at 60°/s and 300°/s. Knee extension/flexion (concentric/concentric muscle action) strength was measured on an isokinetic dynamometer Biodex System II (Biodex Medical Systems, Shirley, NY, USA). The test consisted of 5 and 15 knee flexion/extension repetitions for each leg at 60°/s and 300°/s, respectively [26]. These tests have good reliability and reproducibility [27].

### 2.8. Muscular Imbalance and Asymmetries

Muscle strength imbalances were obtained through the conventional H:Q ratio and the calculation of contralateral deficit between limbs. To assess muscular imbalances in the knees, the conventional percentage of the H:Q ratio was assessed [28]. The asymmetries (contralateral deficit) were obtained through the calculation of the percentage differences between DOM and NDOM limbs by the equation [((Strongest − Weakest)/Strongest) × 100%] as previously suggested [29]. The DOM leg was determined as the leg used to kick a ball [30]. All athletes had the right limb as DOM.

### 2.9. Statistical Analysis

Based on previous studies [31,32] considering a mean difference of 8 cm between PRE and POST VJ tests, a standard deviation of 10 cm and an alpha of 0.05, twelve participants were required to reach a power of 80%. The normality of the demographic data was verified using the Shapiro–Wilk test. Due to the exhibition of a normal distribution, the data are presented as the mean and standard deviation (M ± SD). To check mean differences between DOM and NDOM legs in both PRE and POST moments, the paired Student’s *t*-test was applied at a level of significance of 5% (*p* ≤ 0.05). The mean differences were also calculated using Cohen’s d effect sizes (ES) [33], where an ES of 0.2 represented small differences, 0.5 represented moderate differences, and 0.8 represented large differences [34]. No comparisons were carried out between flexors and extensors or between 60°/s and 300°/s.

## 3. Results

The results of the overall VJ performance are described in Figure 1. A significant increase was found both in SJs and CMJs after the long-term training program. The jump height of SJ started at 26.66 ± 3.98 cm at PRE and increased to 30.71 ± 5.84 at POST, whereas CMJ performance was 28.28 ± 4.64 cm at PRE and 31.73 ± 6.5. This increase represents an improvement of 12% (*p* = 0.017; ES = 0.62) and 15% (*p* = 0.004; ES = 0.82) for CMJ and SJ, respectively. No significant differences were found for the unilateral CMJ performed by both DOM and NDOM legs at PRE and POST moments.

Figure 2 shows the PT (panel A) and the H:Q ratio (panel B) responses either at 60°/s and at 300°/s both for DOM and NDOM limbs at PRE and POST moments. The PT of NDOM flexors at 60°/s (Panel A) were significantly lower than their DOM counter pairs both in PRE and POST conditions (4.6%; *p* = 0.048; ES = −0.22 and 6.3%; *p* = 0.037; ES = −0.33), whereas NDOM extensors at 60°/s had lower and significant PT than NDOM only at POST (7.0%; *p* = 0.048; ES = −0.23). Both DOM and NDOM flexors at 60°/s also had a significant improvement in PT at POST in comparison to PRE (6.7%; *p* = 0.031; ES = 0.51 and 5.6%; *p* = 0.037; ES = 0.48, respectively). The PT of NDOM extensors at 300°/s had a significant increase at POST in comparison to PRE (7.9%; *p* = 0.038; ES = 0.27). Finally, the NDOM at 300°/s had a H:Q ratio (Panel B) significantly higher than DOM both in PRE and POST (8.6%; *p* = 0.041; ES = 0.37 and 11.6%; *p* = 0.013; ES = 0.71, respectively).

The asymmetries found for the CMJ, PT and H:Q ratio are described in Figure 3 (panels A, B and C, respectively). Significant and high asymmetry was found for CMJ at POST in comparison to PRE (panel A), exceeding the threshold of 10%, whereas the PT asymmetries showed any significant difference either for flexors and extensors and for 60°/s or 300°/s. However, similar to CMJ, the PT asymmetries of flexors at 300°/s also outpaced the 10% both at PRE and POST moments.

## 4. Discussion

The present study aimed to examine the fluctuations in VJ and ISK performance and asymmetries throughout 15 weeks of a competitive season in young female volleyball athletes. To the best of our knowledge, no previous studies have analysed the effect of a long-term training program on VJ and ISK using two angular velocities in young female volleyball players. Thus, the current study is pioneering in showing that neuromuscular performance increased and revealed the benefits of 15 weeks of a training program throughout a competitive season. However, in contrast to our hypothesis, some asymmetries also increased after the long-term training program. Then, our main result was to demonstrate that the increase found in the lower limb muscle strength and power seems to occur concomitantly with a certain upper risk of injury, especially for the knees.

The VJ performances found in the present study were consistent with those described in the literature, which report SJ and CMJ performances of female athletes ranging from 20 cm to 32 cm for SJ and from 24 cm to 34 cm for CMJ [31,32,35,36]. The increase found by 12% for CMJ and 15% for SJ throughout the season indicates the positive effect of the training and corroborates previous findings in other team sports female athletes. For instance, Bouteraa et al. [35] described gains of 10% in SJ and 7.5% in CMJ after 8 weeks of balance and plyometric training compared to regular in-season basketball training in female basketball players. Ramirez-Campillo et al. [37] reported significant improvement in jumping ability (SJ = 5.1% and CMJ = 4.4%) after 6 weeks of plyometric training in female soccer players. Häkkinen et al. reported significant increases in a physical fitness profile (SJ = 11% and CMJ = 5%) after 22 weeks of an official competitive season involving 1–2 sessions per week of explosive strength training in female basketball players [38]. However, in female volleyball athletes, the results are conflicting because some authors have described positive effects, whereas others have found impairment or any effect on performance throughout the season. Myer et al., for example, [39] found an increase of 8.7% in VJ performance 6 weeks after a training program that included plyometric, core strengthening, balance, resistance and speed training in young female team sports athletes, including volleyball players. In a similar sample to that used in the present study, after 17 weeks of mixed training sessions, including technical-tactical sessions, aquatic plyometric training and resistance training, Dell’Antonio et al. also found changes near 8% and 14% in CMJ and SJ, respectively [31]. Marques et al. described an improvement of 4% in CMJ after a 12-week training program consisting of resistance training and plyometric exercises in senior female professional volleyball players [32]. Rousanoglou et al. reported an increase of 9.5% in CMJ after a 16-week mixing training schedule involving either general and specific physical conditioning and technical-tactical skills in U-19 women volleyball players [40]. The highest positive training effect was reported by González-Ravé et al., who investigated the seasonal changes in jump performance of female volleyball players and found an increase of 20% in both SJ and CMJ [41]. On the other hand, Bazyler et al. found no statistical changes in SJ in collegiate volleyball players after a 15-week period of a block periodisation model that comprised strength, strength-speed, speed-strength, and a taper. For these authors, the strength volume applied throughout the season was insufficient to produce increases in VJ performance [42]. Newton et al. investigated the effect of 11 weeks of periodized traditional and ballistic resistance training on the strength of the lower limbs in female volleyball players. They found a 5.4% decrease in VJ performance from the start of season to midseason (traditional periodization) and a 5.3% increase from the midseason to end of season (ballistic training period). Therefore, the VJ did not change significantly between the start and end of the season [36]. The divergent results found in the jump height among the abovementioned studies may be explained by the initial fitness status of the participants, the competitive level of the players and the contents of each training program. Indeed, female volleyball athletes seem to show different responses during long-term training comprising general and specific physical fitness and technical tactical contents. Overall, heavy resistance training appears to reduce the neuromuscular responses, whereas reduction of heavy resistance training of the leg extensors and ballistic and plyometric jumps stimulated better neuromuscular performance. Hence, female volleyball players can improve strength and power during the competition season by implementing a well-designed training program that includes both resistance and plyometric exercises [32].

PT represents the maximum torque recorded during muscle contractions at slow or fast angular velocities and is strongly influenced by acceleration and deceleration of the limb during the actions of flexor or extensor muscles [43]. In the present study, the PT performance ranged from 70 N.m^−1^ found in the PRE in the DOM flexors at 300°/s to 205 N.m^−1^ found in the PRE in the DOM extensors at 60°/s. Overall, the results found for PT in the present study were higher than those found in other female volleyball athletes [39,44,45]. Pelegrinelli et al., using a similar ISK test protocol to that used in the present study, found PT values ranging from 65 N.m^−1^ in the NDOM flexors at 300°/s to 141 N.m^−1^ found in the DOM extensors at 60°/s in a sample similar to those investigated in the present study (i.e., U-17 young female volleyball athletes) [46]. The higher ISK performance found in the present study may be explained by the competitive level and the training status of the athletes. Indeed, our data were collected at the start of the preparatory period, whereas Pelegrinelli et al. evaluated their athletes during the preseason (earliest than the preparatory period) [46]. Overall, the studies suggest that elite female athletes had lower ISK performance than male athletes but higher ISK performance than their nonathletic and young female counter pairs [47]. Additionally, ISK performance also depends on many factors, such as age, sex, competitive level, training status, and previous injury.

The effect of long-term training on ISK performance occurred at different magnitudes depending on leg dominance (DOM vs. NDOM), muscle function (flexors vs. extensors) and angular velocities (60°/s vs. 300°/s). In the present study, comparisons between PRE and POST conditions revealed an increase in PT for both DOM and NDOM flexors at 60°/s, whereas only the NDOM extensors showed a significant improvement at 300°/s. In addition, the increase in the PT found in the present study ranged from 5.6% (DOM flexors at 60°/s) to 7.9% (NDOM extensors at 300°/s). In the attempt to compare our results with those of other studies, we found conflicting results about the fluctuations in ISK performance during the season. Some studies describe the weakest (<10%) or no significant changes in ISK performance after a training program [48]. Others report moderate to great improvements (>10%) [49,50,51]. Specifically, in comparison with adolescent female athletes, there are no similar studies with which to make a strong comparison. To the best of our knowledge, only one study has investigated the seasonal changes in knee muscle strength in a team of U-19 female volleyball players [40]. In this study, the authors investigated the effect of 16 weeks on the PT both in flexor and extensor muscles at 60°/s, 180°/s and 240°/s from preparation to a competitive period. With the exception of the increase found in the PT of flexors at 60°/s (approximately 10%), no other significant increase was found. In our opinion, at least two main factors may explain the differences found in our study compared with the others. The first is the teams’ characteristics since the PT flexors at 60°/s found in our athletes at PRE were higher than those found in the others at POST. Therefore, our athletes probably had a higher neuromuscular condition at PRE and, consequently, a lower trainability because there is a premise that how much more trained an individual is, the less trainable it becomes [52]. In other words, physiological adaptations seem to occur rapidly at the onset of a training program, but the rate of improvement declines as physiological systems become more adapted to the training stress. The second is the training contents since the preparatory and competitive periods, commonly focused on different physical fitness related to injury prevention and sports performance, which may enhance neuromuscular performances in different magnitudes and directions [39,53,54,55]. Overall, it is possible to assume that ISK responses depend on the participants´ training status, leg dominance, and angular velocity used for the ISK assessments. Therefore, comparisons among many studies require caution because ISK variables previously reported tested samples with different characteristics and training periods and used unequal ISK test protocols concerning the number of repetitions, angular velocities and muscle actions. Hence, it is important that future studies mirror the previous and use an adequate experimental design accordingly to the specific purposes (i.e., injury recovery or prevention, training effect, intra- or intergroup comparisons) to make better comparisons and conclusions.

In volleyball players, a balanced muscle strength ratio between the agonist and antagonist muscle groups is very important for lower extremity stability and preventing knee injuries [28]. For the optimal functioning of the knee, the quadriceps and hamstrings must work together and balance each other [56]. The activity of the hamstring muscles is balanced by the quadriceps muscle, which has an average strength 50% to 100% greater than that of the hamstring muscle [47]. Unopposed quadriceps muscle contractions can translate the tibia anteriorly and cause significant strain in the anterior cruciate ligament (ACL) [57]. Therefore, the balance of power between the quadriceps and hamstring muscles is crucial to normal knee function because the quadriceps muscles can produce forces in excess of those needed for ligament tensile failure [47]. The conventional H:Q ratio is defined as the ratio between the peak torque of the hamstring and the quadriceps concentric contraction assessed in ISK tests [28]. Commonly, the H:Q ratio is important because it reflects muscle imbalances, which are related to a higher risk of injury in soft tissues, joints and muscles [58]. The concentric H:Q ratio found in the present study was near 50% at 60°/s and ranged from 73% to 88% at 300°/s. These results suggest that the training program applied during the 15 weeks was probably not enough to reach the H:Q ratio recommendation to prevent injuries in the knee joint. Therefore, an adjustment in the resistance training density with a better balance between intensity and volume should be reorganized for the next seasons. Although no relationship between the H:Q ratio and angular velocities has been suggested for females [59], our results were lower than those found at 60°/s but jibes with the H:Q ratio reported at 300°/s in female athletes of several sports [59]. Devan et al. suggested that athletes with an H:Q ratio of less than 80% at 300°/s had a greater occurrence of overuse knee injuries [60]. According to these authors, an H:Q ratio above the normal ISK range is >69% at 60°/s and >95% at 300°/s and below the normal ISK range is >60% at 60°/s and <80% at 300°/s [60]. In this sense, our athletes seemed to start the competitive season with a lower H:Q ratio and a certain risk of injury. Moreover, we found that NDOM limbs had significantly higher values than DOM limbs in the H:Q ratio at 300°/s both in PRE and POST moments. This result highlights the need to increase preventive programs to avoid joint and muscle injuries throughout the season and agrees with previous findings suggesting that a supervised integrative neuromuscular training program comprising general and specific strength and conditioning activities is able to enhance health and reduce knee pain and other lower limb sports-related injuries in volleyball athletes [8,61]. Overall, it has been recommended that a preventive injury program for youth athletes should include a professional screening, an adequate diagnosis of risks of injuries, and a supervised training program focused on enhancing neuromuscular coordination, joint mobility, flexibility, balance, core strength, and overall general physical fitness performance related to health and sports performance.

The asymmetry reflects the muscle imbalance from one limb to another and may be defined as contralateral deficit [18]. The long-term response of contralateral deficit is important for athletes who perform jumping on one leg, such as volleyball players, because even when the training is conducted correctly, the specific tasks imposed by the sport may develop the DOM body side more than the NDOM, producing a certain degree of asymmetry between limbs [62]. The determination of contralateral deficits in athletes can provide insights related to performance, prevention, rehabilitation and return to play after injury problems [12,14]. Moreover, interlimb asymmetries are associated with decrements in physical performance in youth elite team sports athletes [23]. Previous research has suggested a 10–15% threshold of neuromuscular asymmetry to be normal physiological variability [11,14,15,16]. In the present study, we observed asymmetries ranging between 4.8% in the PT of flexors at 60°/s at PRE and 18.7% in the H:Q ratio at 300°/s at POST. Although only the CMJ asymmetry showed a significant increase after the training program in comparison to PRE (102%), it is important to note that, even without any other significant difference found in all the other asymmetries, the asymmetry of the PT of flexors at 300°/s as well as the H:Q ratio at 300°/s overtook the considered normal physiological threshold for injury risk since the start of the preparatory period season. In addition, they remained at a high risk of injury zone after the training program. These results are in agreement with previous studies that found an increase in asymmetry even after a long-term training program [63,64,65]. However, contralateral asymmetry is a controversial topic in the literature because some studies have described important changes in interlimb asymmetries in young athletes after a specific training program [64,66], while others have found any effect [67,68]. The main reason for the discrepancies found among the studies seems to be the variable chosen to assess contralateral asymmetry because many contralateral neuromuscular asymmetries found throughout a season show different magnitudes and directions of change. For instance, Fort-Vanmeerhaeghe et al. compared interlimb asymmetry magnitudes across the season and found significantly higher asymmetries in the unilateral CMJ at mid-season in comparison with preseason and end-season, but the interlimb asymmetry magnitude of the one-leg hop test remained unchanged throughout the season. With this result, the authors suggest that the unilateral CMJ is the test with a greater sensitivity in detecting the magnitude of asymmetry in youth elite team-sport athletes [69]. Similar results have been reported by Madruga-Parera et al., who found interlimb asymmetry scores ranging from 1.83 to 15.03%, with the unilateral CMJ presenting the greatest magnitude of asymmetry [70]. Complementarily, the authors also found significant differences between interlimb asymmetry scores across multiple tests, reinforcing the recent suggestions that asymmetries are task- and variable-specific [71]. Therefore, comparing asymmetry between different tests, samples and training modes requires caution.

The absence of a reduction in asymmetry found in the present study suggests that the physical training applied during the preparatory period was not enough to normalize the asymmetries found at the beginning of the season. From our point of view, there are at least three main reasons that could explain the increase or the maintenance of the asymmetries. First, although our athletes were young, they also had a relatively higher time of volleyball practices (5 years). Then, it is possible that the contents and the relatively short time used as a preparatory period were not enough to normalize the asymmetries developed by the chronic neuromuscular adaptations that occurred during their full carrier. This evidence has been confirmed by previous authors who showed higher asymmetries in more experienced highly skilled athletes than in their unexperienced less skilled counter pairs [46,72]. Second, the neuromuscular training program was primarily designed to improve physical fitness related to overall volleyball performance, and the reduction in interlimb asymmetries was not prioritized. Although the best training strategy to decrease interlimb differences currently remains unclear, some strategies have been successful in reducing neuromuscular asymmetries. Overall, unilateral strength exercises [73,74], strength and power exercises focusing on the major muscles of the lower body and trunk musculature [68], and a long-term well-structured multifaceted neuromuscular exercise-based injury prevention program with a combination of muscular strength as well as proprioceptive balance and stabilization exercises were likely to have been beneficial to reduce some important interlimb asymmetries related to sports performance and injury risk [75,76,77]. Third, the training sessions in the court with specific volleyball technical skills had a relatively high volume, which could reinforce the existing asymmetries or reduce the possible benefits of the neuromuscular training by the concurrent training effect [78]. Indeed, concurrent strength and endurance training appear to inhibit strength development when compared with strength alone [79]. Moreover, resisted training seems to improve the muscle strength of female volleyball players only if they are not undergoing concurrent training or if the training program is less than 3 days per week [32]. Overall, the recent recommendations to avoid or minimize the interference effect related to concurrent training in team sports athletes are to adequately manipulate the overall factors related to training adaptation, such as exercise orders, frequency, intensity, volume and recovery [80].

Concerning the training program applied in the present study, some important considerations can be highlighted. A positive and significant effect was found on VJ performance, but marginal or insufficient effects were related to the H:Q ratio and asymmetry. This finding is in agreement with recent meta-analytical studies suggesting that weightlifting and plyometric training are better strategies to improve jumping performance [81]. Otherwise, the training characteristics were not able to reduce either the H:Q ratio or asymmetry. This result suggests the necessity of increasing the number of stimuli focused on eccentric strengthening training and stretching to improve muscle functions related to hamstring strength, fascicle length, H/Q ratio, limb asymmetry and flexibility, as recommended for recent meta-analytical studies [82,83,84,85].

While the current study had certain strengths, such as the originality of describing for the first time the long-term asymmetries and performance responses of the VJ and ISK tests at two angular velocities in young female volleyball players, we recognize some limitations that should be mentioned when interpreting the current results. The training load monitoring and intersession recovery status were not controlled. This information could give us a better comprehension to explain why neuromuscular and asymmetry fluctuations occurred. The absence of a control group is another important limitation. This is especially because muscle mass and maximum strength values of female adolescents continue to develop after the onset of menarche. Therefore, we suggest that future studies include a control group and their experimental design. However, it is also important to note that the present study was performed with high-level young athletes, and in practical terms, it would be challenging to recruit volunteers with the same characteristics, training routines and similar performance levels as our athlete sample. The lower sample size also did not allow us to perform an analysis by playing position. It would be helpful to understand how the asymmetries could change according to the specific demands of each position on the court. The absence of the researchers to influence the athletes´ routines and the staff’s decisions is another limitation since a better interaction could optimize some training responses. An eccentric assessment in the ISK test would also provide another perspective of strength. A follow-up throughout the full season may also help to better comprehend whether the asymmetries found may or may not be related to injuries. Finally, the menstrual cycle was also not considered during the tests, and some interpretations may be limited because neuromuscular performance can be affected differently according to the follicular or luteal phases of the menstrual cycle. Future studies may be designed to fill these gaps in the literature.

## 5. Conclusions

Our results demonstrate that 15 weeks of a training program comprising preventive injury and neuromuscular and technical-tactical exercises were able to improve the physical fitness related to volleyball performance in elite young female athletes. However, asymmetries increased concurrently with the VJ and ISK performance in different magnitudes. These findings can provide valuable information to assist physiotherapists, coaches and physical conditioners in the prescription of exercises to reduce muscle asymmetries, prevent injury, and improve lower limb muscle strength and power performance according to the individual aims and conditions related to each specific demand.

## Figures and Tables

**Figure 1 ijerph-19-16420-f001:**
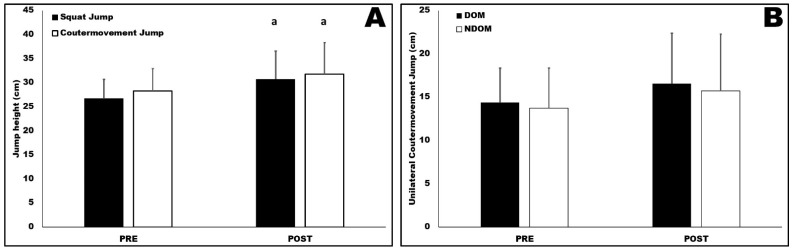
Jump height of bilateral squat jump and countermovement jump performances (**A**) and unilateral countermovement jump performance at the start (PRE) and after 15 weeks of a competitive period (POST) (**B**). DOM = dominant leg; NDOM = nondominant leg; a = different than PRE (*p* < 0.05); bilateral squat jump and countermovement jump were not compared.

**Figure 2 ijerph-19-16420-f002:**
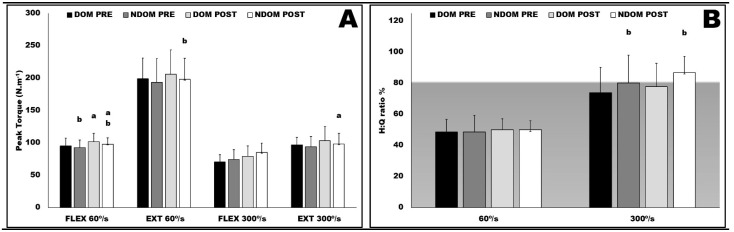
Peak torque (**A**) and H:Q ratio (%) (**B**) at 60°/s and at 300°/s either for DOM and NDOM limbs and both in PRE and POST moments. FLEX = flexors; EXT = extensors; DOM = dominant leg; NDOM = nondominant leg; PRE = prior to the in-season starting; POST = after 15 weeks of a competitive period; a = different than DOM at the same period (*p* ≤ 0.05); b = different than PRE for the same leg or muscle group (*p* ≤ 0.05). The gray area (<75%) indicates an injury risk zone.

**Figure 3 ijerph-19-16420-f003:**
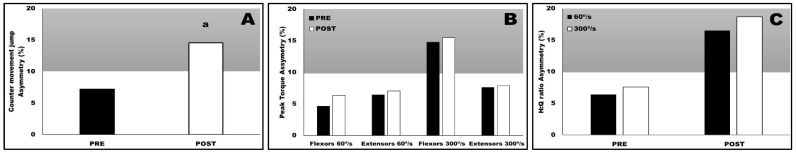
Asymmetry of unilateral CMJ (**A**) and ISK variables (peak torque, total work and average power = (**B**)) and H:Q ratio (**C**) at 60°/s and 300°/s in both PRE and POST conditions. DOM = dominant leg; NDOM = nondominant leg; PRE = prior to the competitive season starting; POST = after 15 weeks of a competitive season; a = different than DOM at the same period (*p* ≤ 0.05); The gray area (>10%) indicates an injury risk zone.

**Table 1 ijerph-19-16420-t001:** The training schedule comprised the number of each training session at the gym and at the volleyball court (technical-tactical) and the training volume of the weekly training throughout the 15-week preparatory and competitive periods. The black square indicates weeks with official volleyball matches. GC = general conditioning. STRE = strength. POW = power; PLYO) = plyometric.

Training Program																
Training Type	Week	Basic			Preparatory					Pré Competitive				Competitive		
		1	2	3	4	5	6	7	8	9	10	11	12	13	14	15
**Resistence Training**	**Main Goals**	GC	GC	Stre	Stre/Pow	Pow	Plyo	Stre/Pow	Pow	Stre/Pow	Plyo	Stre/Pow	Plyo	Pow	Stre	Stre/Pow
	**Sets × Reps**	3 × 15	3 × 12	3 × 4	2 × 4 3 × 6	3 × 4 3 × 6	4 × 6	2 × 4 3 × 6	3 × 6	2 × 4 3 × 6	4 × 6	2 × 4 3 × 6	4 × 6	3 × 6	4 × 4	4 × 4
	**Rest (min)**	1	1	2–3	2–3	3	3	2–3	5	3–5	3	3–5	3	5	3	3–5
	**Load**	60–70% (1RM)	60–70% (1RM)	60–90% (1RM)	60–90% (1RM)	60% (1RM)	Body weight	60–90% (1RM)	90% (1RM)	80% (1RM)	Body weight	80% (1RM)	Body weight	80% (1RM)	90% (1RM)	70%/80% (1RM)
	**Sessions**	5	5	5	5	5	5	5	5	4	5	4	5	4	3	4
	**Volume (min)**	390	540	390	420	420	600	420	360	330	600	240	600	210	210	240
**On court**	**Main Goals**	Development of technical skills (defense, reception and service) and tactics														
	**Sessions**	5	5	5	5	5	5	5	5	5	5	5	5	4	4	4
	**Volume (min)**	570	570	570	570	570	570	570	570	570	570	570	570	510	465	510

**Table 2 ijerph-19-16420-t002:** The training program performed during 15 weeks of the season.

	Period	Basic	Prep	Pre-Comp	Comp
	Weeks	1–3	4–8	9–12	13–15
	Exercises	Sets × Reps (RM)			
**Upper Limbs**	Alternating dumbbell curl	4 × 10	4 × 8	3 × 8	3 × 6
	Crossover crucifix	4 × 10	4 × 8	3 × 8	3 × 6
	Inverted crucifix machine	4 × 10	4 × 8	3 × 8	3 × 6
	Dumbbell shoulder press	4 × 10	4 × 8	3 × 8	3 × 6
	Vertical pull	4 × 10	4 × 8	3 × 8	3 × 6
	Ball dumbbell press	4 × 10	4 × 8	3 × 8	3 × 6
	Medicine ball crunch throw	4 × 10	4 × 8	3 × 8	3 × 6
	Wrist flexion and extension	4 × 10	4 × 8	3 × 8	3 × 6
	Triceps pulldown	4 × 10	4 × 8	3 × 8	3 × 6
	Triceps rope pulldown	4 × 10	4 × 8	3 × 8	3 × 6
	Pull over	4 × 10	4 × 8	3 × 8	3 × 6
	Bench press	4 × 10	4 × 8	3 × 8	3 × 6
	Dumbbell rows	4 × 10	4 × 8	3 × 8	3 × 6
	Incline dumbbell fly	4 × 10	4 × 8	3 × 8	3 × 6
**Lower limbs**	Calf machine	4 × 10	4 × 8	3 × 8	3 × 6
	Anterior tibial pulley	4 × 10	4 × 8	3 × 8	3 × 6
	Sumo squat	4 × 10	4 × 8	3 × 8	3 × 6
	Step up	4 × 10	4 × 8	3 × 8	3 × 6
	Side step	4 × 10	4 × 8	3 × 8	3 × 6
	Leg press 45°	4 × 10	4 × 8	3 × 8	3 × 6
	Olympic lifting	4 × 10	4 × 8	3 × 8	3 × 6
	Kettlebell single leg squat	4 × 10	4 × 8	3 × 8	3 × 6
	Kettlebell squat	4 × 10	4 × 8	3 × 8	3 × 6
	Swiss ball wall squat	3 × 30″	3 × 30″	3 × 30″	3 × 30″
	Seated leg curl	4 × 10	4 × 8	3 × 8	3 × 6
	Unilateral seated leg curl	4 × 10	4 × 8	3 × 8	3 × 6
	Unilateral leg curl	4 × 10	4 × 8	3 × 8	3 × 6
	Prone leg curl	4 × 10	4 × 8	3 × 8	3 × 6
	Back squat	4 × 10	4 × 8	3 × 8	3 × 6
	Abductor & adductor machine	4 × 10	4 × 8	3 × 8	3 × 6
**Core**	Abdominal exercices	3 × 40	3 × 45′	3 × 60	
	Ankle bosu ball stability	3 × 40″	3 × 45″	2 × 60″	
	Knee bosu ball stability	3 × 40″	3 × 45″	2 × 60″	
	Battle ropes on the bosu	3 × 40″	3 × 45″	2 × 60"	
**Plyometric**	Depth jump	4 × 6	4 × 8	3 × 6	
	Ninja landings	4 × 6	4 × 8	3 × 6	
	Basic box jump	4 × 6	4 × 8	3 × 6	
	Depth to broad	4 × 6	4 × 8	3 × 6	
	Lateral rop	4 × 6	4 × 8	3 × 6	

## Data Availability

The data presented in this study are available on request from the corresponding author. The data are not publicly available due to ethical factors and the privacity of the players.

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
