# Peer review of "Long-Term Changes in Vertical Jump, H:Q Ratio and Interlimb Asymmetries in Young Female Volleyball Athletes"

_ijerph, 2022, doi:10.3390/ijerph192416420_

Round 1

Reviewer 1 Report

Abstract:

Neuromuscular changes are scarce?  This is quite broad and non-direct. This topic sentence does not directly relate to the aim of the study.

Fluctuations in what?

State the p values

Introduction:

Line 49-50: You said elicit twice in this sentence.

Line 61: male – proofread. This sentence reads very awkward.  It is very hard to understand.

I am not sure how time availability relates to this purpose. Or anaerobic and aerobic contributions. Is the aim to determine neuromuscular capabilities? That is how your abstract read.

Strength to time to neuromuscular capabilities in two sentences without explanation.  I am having a difficult time following the topic here.

Neuromuscular is used to loosely in the introduction with no definition are explanation towards the study.  Need to give information about sensory and motor control and how it relates to you your variables, otherwise just discuss muscle activation and asymmetries.  You are not directly measuring neuromuscular capabilities.

More detail in neuromuscular training needs to be of focus in the intro.

Methods:

How was adherence measured?

What is the equation used for asymmetry?

Reviewer 3 Report

INTRODUCTION

In the opinion of this reviewer, the introduction does not have a clear line of argument.

Volleyball is one of the most popular sports in the world, both individual and collective; and both collective opposition and unopposed. If the authors are going to allude to the popularity of volleyball, it would be desirable to have some kind of data regarding the number of worldwide licenses or reference that supports this statement. 

In the opinion of this reviewer, the paragraph briefly explaining the regulation does not provide any relevant information (lines 35 to 42) and should be deleted. In addition, aspects such as the 5th set is played at 15 points are explained, but the differentiated score at which the remaining 4 sets are played is not explained.

Line 42 and 43 -  The statement should incorporate reference. This reference can be found on line 48. Estimates of working and rest time should help justify that this is an intermittent sport. Include the competition. But beware, the high-level sport has more rest times due to differentiated regulatory elements such as technical times, the Challenge System, or the break added at the end of the 2nd SET. These breaks do not exist in the Under 17 category. 

Line 48 – Different authors have classified the basic technical skills in volleyball calling them technical actions, technical skills or main skills of volleyball. I think it is convenient to cite one of these classifications and change the name of motor tasks.

The article relates to strength in volleyball. However, the most relevant manifestations of strength in volleyball are not explained. This aspect is relevant, as the intervention should be related to the specific needs of volleyball.

Line 53 – Volleyball is a sport with unilateral hits in serve and spike actions. In addition, these actions are executed at high speed, being the power used in their use, a variable related to their performance. So why do authors only name actions related to the lower limb? If the article only studies the lower limb, the title is not consistent with the study.

Línea 59  Developing muscle strength: why is especially important in female volleyball athletes?   It´s not clear why is more important than in men´s volleyball.

Paragraphs 69-78. The authors present vague general recommendations for improving or maintaining strength levels based on the indications of a single study and without incorporating their training methods. This reviewer does not find this paragraph of clear use for the introduction of the topic. I suggest its removal. 

Line 95 What test of vertical jump?  single leg countermovement jump? countermovement jump?  Not all vertical jumps are recommended to measure asymmetries. It must be justified in method.

2. Materials and Methods

In general, this reviewer finds important doubts that affect the validity of the results in the study methodology. Especially the absence of a group control .

Line 129 - The players are defined as defining the group as national-level elite U-17. How do you define elite players? At what level do the participants play? Information from years of experience in practice is less relevant than the level of competition. The average height does not seem to reflect an elite group of this category. Nor do the results of the jumping tests reflect an elite under-17 team.

In addition, the previous training of the athletes in strength is a relevant aspect to understand the effect that the program had.

Line 138 - Why did the authors choose the SJ and CMJ tests?  Is there literature that supports the use of these tests as suitable to measure imbalances, asymmetries and performances? Review this aspect. As an example, Bishop et al. (2018), justify the use of unilateral or bilateral tests.

Bishop, C., Read, P., Lake, J., Chavda, S., & Turner, A. (2018). Interlimb asymmetries: Understanding how to calculate differences from bilateral and unilateral tests. Strength & Conditioning Journal40(4), 1-6.

The tests to be carried out are not clearly explained: first we talk about CMJ, then we include in line 187 unilateral CMJ. They must clarify this. 

2.4. Training Schedule

This section presents serious problems in the opinion of this reviewer.

Lines 148-159 - The specific work of gym is widely used by physical trainers to reduce imbalances and asymetries avoiding injuries, in addition obviously to improve muscle performance. Therefore, the results of this work will depend in part on the exercises used in the proposed physical preparation. It does not explain how many exercises are going to be performed. Those exercises not are explained in the methodology. Based on this, replicability of the study with a different sample is not possible.

What tipe of of multijoint exercises have been used? How have the load variables of these exercises been calculated?

Line 167 – Missing reference.

Line 168 - What measuring instrument was used?; Does it have validation in its measure?

Line 186 - Why was only the best mark chosen and not an average of the 3 jumps? I recommend reviewing the following article:

Bishop, C., Read, P., Chavda, S., Jarvis, P., & Turner, A. (2019). Using unilateral strength, power and reactive strength tests to detect the magnitude and direction of asymmetry: A test-retest design. Sports7(3), 58.

2.9. Statistical Analysis

Line 215 – Cite which are the previous studies. However, previous studies have used a different sample. Therefore, the average differences of previous studies cannot be assimilated. In addition to calculating significance, I consider it desirable to use an effect size meter.

RESULTS

The program shows aconsiderable increase in jump height for 15 weeks of training with increments of 12% and 15%. If the study was developed after a prolonged period of vacation rest, this aspect could affect the improvement produced in these weeks of training. Technical training on the track after a period of inactivity can be a sufficient stimulus to improve performance in the jump. They should clarify this aspect. Therefore, it would be difficult in the opinion of this reviewer, to establish which is the effect of strength training in the gym and which of the stimulus of technical training. Therefore, in the opinion of this reviewer, it would be absolutely necessary in the design of the study to include a control group, that only carried out the technical-tactical training, to verify the improvement obtained thanks to the program.

Discussion

I understand the relevance in this case of the orientation of the work towards asymmetries. But the fact that they do not have a control group, raises doubts in the attribution of improvements to the strength training program. Especially in stages of development, in which physiological maturation and technical training, can produce improvements in strength levels. In fact, the results obtained are superior of most of the studies that have been reviewed.  Based on this, the validity of the discussion may raise doubts.

Round 2

Reviewer 3 Report

NTRODUCTION

In the opinion of this reviewer, the introduction has been significantly improved, now presenting a greater plot coherence. I think the removal of some paragraphs has been positive.

METHODOLOGY

Improvements have been introduced in the identification of the exercises. However, there are still some unclear exercises: examples: medicine ball toss:: how did they throw it? This clarification is essential to understand the muscles involved and the possible transfer with the kinematics of volleyball; box jump: height and type of jump?. The range of work percentages is still very wide and progression is not detailed. It is not the same to work at 30% than at 60% of a maximum repetition.

I agree with the authors that the onset of menarche is a very relevant physiological event in the development of strength. However, muscle mass and maximum strength values ​​continue to develop after the onset of menarche. The mean age of the sample suggests that it is still in the process of development (15.34±1.19 years). Therefore, it is a variable that is a limitation of the study; especially since there is no control group.

Thus, the main aspect in the opinion of this reviewer is the control group. As the authors indicate, there are published articles that have not included a control group, using a methodological aspect typical of quasi-experimental methodology. However, in the opinion of this reviewer, it is a critical aspect of the work. I reiterate the idea that a control group is essential to be able to determine the association of the training carried out on the observed improvements. Otherwise, it is impossible to determine the improvement that technical-tactical training has produced, incorporating repeated actions related to power at maximum and sub-maximal intensities, on young people in a sensitive stage of strength development.

Finally, review the values ​​taken as reference in the calculation of d Cohen:

Cohen, J. (1992). A power primer. Psychological bulletin, 112(1), 155-159.

https://doi.org/10.1037/0033-2909.112.1.155
